# Identification and Genetic Diversity of *Spodoptera frugiperda* J. E. Smith (Lepidoptera: Noctuidae) in Egypt

Kreema A. El Lebody [1], Rasha G. Salim [2], Ghada M. El-Sayed [2] and Shaymaa H. Mahmoud [3,*]

[1] Plant Protection Research Institute, Agriculture Research Centre, Dokki, Giza 12627, Egypt
[2] Microbial Genetic Department, Biotechnology Research Institute, National Research Centre, 33 El-Bohouth St. (Former El-Tahrir St.), Dokki, Cairo 12622, Egypt; rasha_gomma@yahoo.com (R.G.S.); ghada.khalefa@yahoo.com (G.M.E.-S.)
[3] Zoology Department, Faculty of Science, Menoufia University, Shibin El Kom 32511, Egypt
[*] Correspondence: drshaymaahussein@gmail.com

**Abstract:** Fall armyworm, *Spodoptera frugiperda*, is a serious agricultural pest native to tropical and subtropical regions of the Western Hemisphere and has invaded Africa and further spread into most countries of Asia within two years. In Egypt, researchers have conducted thorough investigations into the behavior of the fall armyworm and various methods to manage its impact. This study aims to further our understanding of the genetic diversity of fall armyworm populations in Egypt. By collecting specimens from five provinces across the country, we sought to analyze their genetic makeup. Through examination of partial sequences of the mitochondrial cytochrome oxidase subunit I (COI), we identified three distinct haplotypes. Phylogenetic analysis suggests that the primary sources of *Spodoptera frugiperda* in Egypt likely stem from both Asian and African origins. Employing the PCR-RFLP technique on the complete COI sequence, we were able to discern genotype patterns within the fall armyworm population. Our findings indicate the presence of two distinct strains—the Corn and Rice strains—within Egypt. This research contributes essential insights into the genetic diversity of insects in Egypt, providing valuable knowledge that can inform more effective strategies for managing this agricultural pest.

**Keywords:** *Spodoptera frugiperda*; fall armyworm; COX I; RFLP; phylogeny; genetic diversity; invasive species

## 1. Introduction

Global food security is threatened by various factors, including insect pests, diseases, extreme weather, and rapid population growth [1]. Among these challenges, the fall armyworm (FAW), scientifically known as *Spodoptera frugiperda* (J.E. Smith) (FAW, Lepidoptera: Noctuidae: Noctuinae), poses a significant pre-harvest threat to agriculture in sub-Saharan Africa [2]. The FAW is highly polyphagous, has a particular affinity for corn (Poaceae), and its larval stage can be highly destructive, potentially leading to plant death [3]. In 2016, the fall armyworm (FAW) first appeared in Africa, starting in Nigeria and swiftly spreading to over 28 countries in Southern and Eastern Africa, causing significant damage to corn crops, with losses exceeding 70%, and to a lesser extent, to sorghum and other crops [4]. Subsequently, in May 2019, the fall armyworm (FAW) was initially detected in corn fields in a village located in Kom-Ombo City, Aswan. This village is positioned on the eastern bank of the Nile River, in the southernmost region of Upper Egypt, as confirmed by the Agricultural Pesticide Committee (APC) of the Ministry of Agriculture. It later spread to Luxor, Qena, and Sohag governorates, all situated on the west bank of the River Nile in the southern region of Egypt. By 2021, it had also invaded the Assuit governorate (located in the southern part of Egypt), causing damage to the Sorghum fields [5]. However, extensive surveys of the genetic and biological characterization of FAW that infested Egypt need to be investigated and documented for further management of this biological risk.

Genetic analysis has provided valuable insights into the origins, preferred crops, and susceptibility of fall armyworm (FAW) to insecticides. *S. frugiperda* encompasses two distinct genetic strains: the 'Rice strain' (R strain) and the 'Corn strain' (C strain), each exhibiting different preferences for hosts. The R strain is commonly found in millet and grass species, whereas the C strain tends to favor corn, cotton, and sorghum [6]. Furthermore, studies have indicated variations in mating behaviors and the composition of sex pheromones between these strains. Although there is a correspondence between genetic markers and host plants, it is not absolute, with approximately 20% of larvae from corn expressing RS markers on average and sporadic instances of more significant discrepancies [7]. As the genetic structure variation of a pest population in both space and time, along with gene flow among its subpopulations, significantly influences the rate of resistance evolution, estimating gene flow using molecular markers can serve as an index of the dispersal rate, and molecular markers have been widely employed to assess genetic similarity and estimate gene flow among insect populations [8]. Therefore, understanding insect population genetic structures, intraspecific gene flow, and biological aspects is crucial for devising management strategies to delay the evolution of resistance to control methods [9]. Several molecular markers have been used to discriminate between strains of *S. frugiperda*—the "R" and "C" strains. For instance, mitochondrial DNA (mtDNA) analysis is generally considered more powerful than allozyme analysis for revealing population structure and has been employed in numerous population genetic studies. Variations in the cytochrome oxidase I (COXI) and cytochrome oxidase II (COXII) regions of the mtDNA genome have proven useful for measuring genetic variation in various insect taxa, particularly when compared to nuclear DNA variation [10]. Constructing a phylogenetic tree based on the nucleotide sequences of COXI provides a reliable indication of the biological origin and genetic flow of insects.

The PCR-RFLP technique, which stands for restriction fragment length polymorphism, is a precise technique and reasonably priced way to identify different species of insects, applicable from egg to adult developmental stages. Consequently, it demonstrated reliable potential for distinguishing between both strains of *S. frugiperda* [11].

In this study, we present the first report investigating the genetic diversity of fall armyworms collected from different governorates throughout Egypt, both in Upper and Lower Egypt. The genetic diversity included the identification of collected strains by nucleotide sequencing of the COXI gene, which was then compared with those deposited in GenBank. Moreover, this study focused on determining whether there are both strains (C-strain and R-strain) in Egypt and their distribution using the PCR-RFLP technique. A strain known as the KRHOSSAM strain underwent twelve generations of cultivation in a laboratory setting. This strain served as a reference model to explore potential variations in genetic diversity and polymorphism between laboratory and field conditions. Additionally, it aimed to discern whether these variations were influenced by environmental compatibility when compared to other strains. Finally, we concluded whether the use of COXI and PCR-RFLP investigations performed as anticipated.

## 2. Materials and Methods

### 2.1. Insect Collection

Insect collection was performed from corn crops that were clearly exhibiting infestation symptoms and severe damage on corn leaves, with several holes and jagged edges. Upon deeper examination, larvae linked to feeding in the funnels were discovered (Figure 1). Insect specimens were gathered from various locations across Upper and Lower Egypt between 2022 and 2023. These locations include Aswan, Sohag, Qalyubia, Menoufia, and Kafr el-Sheikh, as illustrated in Figure 2. Aswan and Sohag represent Upper Egypt, while Menoufia, Qalyubia, and Kafr el-Sheikh represent Lower Egypt. Primarily collected from early summer corn fields in August, additional specimens were obtained from Nile corn fields in October in Kafr el-Sheikh. Each population was named according to the governorate from which it originated. Additionally, a population was established from egg

masses gathered from a corn field in Aswan and subsequently reared in the laboratory for 12 generations. This laboratory-reared strain served as a reference model for comparing genetic configurations with those subject to natural selection and migration, analyzed through the COXI gene nucleotide sequence and PCR-RFLP pattern. From each governorate, one insect specimen comprising ten larvae at the 5th instar was selected for COXI characterization and investigation of polymorphism. However, in Kafr el-Sheikh, two specimens were chosen for the same purpose.

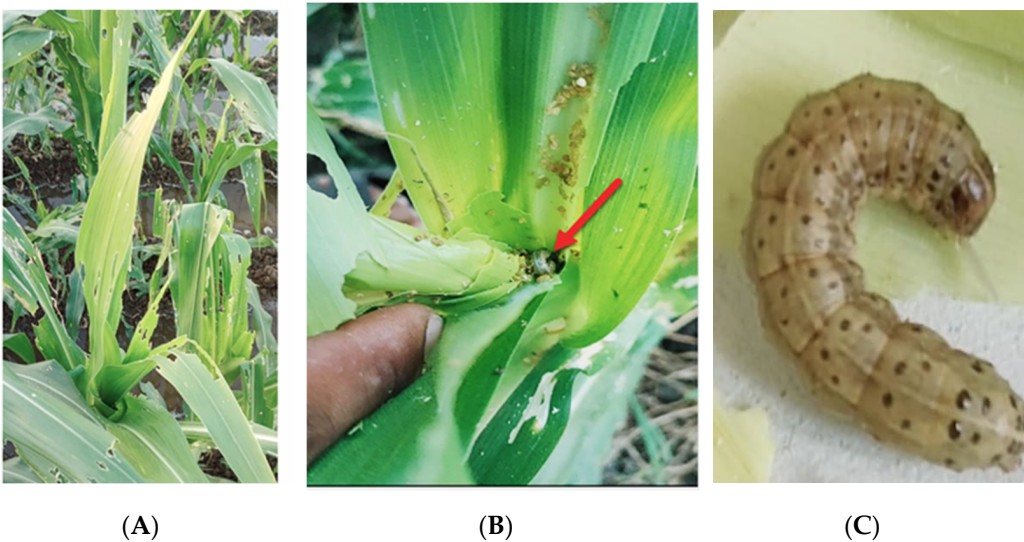

(**A**)    (**B**)    (**C**)

**Figure 1.** Maize-field damage observations caused by *S. frugipedra* larvae. (**A**) Severe damage to maize leaves with several holes and jagged edges. (**B**) Feeding funnels in corn leaves, with the red-colored arrow indicating the larvae. (**C**) The fall armyworm larvae exhibit four black spots on their second abdominal segment, four small black dots on each subsequent segment, and an inverted white Y-shape on their head capsule.

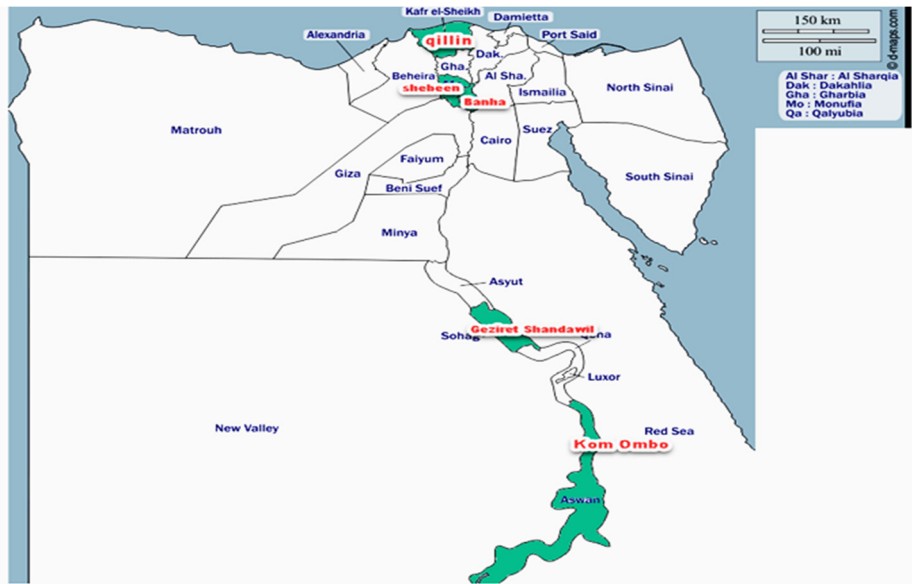

**Figure 2.** Map of Egypt pinpointing the five specific locations where *S. frugiperda* larvae were collected. Each location corresponds to a governorate: Kom Ombo corresponds to Aswan, Gezeiret Shandawil represents Sohag, Benha represents Qalyubia, Shebeen represents Menoufia, and Qillin represents Kafr el-Sheikh.

All specimens were brought to the Laboratory of Microbial Genetics at the National Research Centre in Egypt by preserving them in a single sterile micro-centrifuge tube filled with 70% ethanol, as described by Acharya et al. [12].

### 2.2. Molecular Identification

#### 2.2.1. Genomic DNA Extraction, PCR Amplification, and Sequencing of mtCOXI Gene

Genomic DNA was extracted from each larva using a genomic DNA kit (Gene JET DNA Thermo Scientific$^{TM}$, Waltham, MA, USA) according to manufacturer instructions. Molecular identification relies on the sequence characterization of the mitochondrial cytochrome c oxidase subunit I (COXI) gene. PCR amplification was initiated using a pair of universal primers, LCO1490 (5′-GGTCAACAAATCATAAAGATATTGG-3′) and HCO2198 (5′-TAAACTTCAGGGTGACCAAAAAATCA-3′) as described by [12], to amplify a specific COXI gene fragment ranging from 600 to 700 bp. In PCR tubes, a 50 μL reaction mixture was prepared as follows: 2 μL of each primer (forward and reverse) at a concentration of 10 pmol, 25 μL of 2x EmeraldAmp$^®$ GT PCR Master Mix from Takara Bio (Kusatsu, Japan), 4 μL of genomic DNA template, and adjusted to 50 μL with PCR-grade water. The PCR process involved an initial denaturation at 95 °C for 3 min, followed by 35 cycles of denaturation at 95 °C for 30 s, annealing at 48 °C for 30 s, and elongation at 72 °C for 1 min. The final extension was carried out for an additional 10 min at 72 °C using the BioRad T100 Thermal Cycler (Bio-Rad Laboratories, Singapore). The amplified fragment was analyzed on a 1.2% agarose gel stained with ethidium bromide. Identification of molecular weight was guided by a 1 kb DNA ladder (Thermo Scientific™, Waltham, MA, USA). PCR products were purified using the GeneJET Gel Extraction Kit from Thermo Scientific™ (Waltham, MA, USA). Gel visualization and photography were performed using the XR+ Gel Documentation System (Bio-Rad, Hercules, CA, USA). The purified PCR product underwent Sanger sequencing using the sequencer 3500 genetic analyzer and the big dye X terminator kit (Thermo Fisher, Waltham, MA, USA) for both forward and reverse directions.

#### 2.2.2. Sequence Analysis and Phylogeny Construction

The resulting sequences were edited using BioEdit 7.1.10 software [13] and were compared to those in the GenBank database (http://www.ncbi.nlm.nih.gov/blast, accessed on 10 December 2022). A set of 23 COX1 genes from the GenBank database of related species was utilized to construct a phylogenetic tree, in addition to the COXI gene of *S. litura* as an outgroup. The entries standing for COXI gene sequences isolated from the continents of Africa, Asia, and America (North and South) are listed in Table 1. Clustal W [14] was utilized for nucleotide sequence alignment. Utilizing the neighbor-joining method, we inferred the evolutionary history [15] and generated an optimal tree. Alongside the branches, we indicated the proportion of duplicate trees where related taxa clustered together in the bootstrap test (1000 replicates) [16]. The tree was drawn to scale, with branch lengths reflecting the evolutionary distances estimated for the phylogenetic tree. These distances, measured in base differences per site, were calculated using the p-distance method [17]. This analysis comprised 33 nucleotide sequences. To conduct evolutionary studies, MEGA11 was employed after eliminating any ambiguous locations for each sequencing pair through pairwise deletion [18].

**Table 1.** Accession numbers assigned to COXI genes of *S. frugipedra* isolates in different countries.

| Country | COXI Accession Numbers | Continent |
| --- | --- | --- |
| Bangladesh | MT933063.1 | Asia |
| Bhutan | MT324066.1 | Asia |
| Brazil | JF854740.1 | South America |
| Burkina Faso | MT152769.1 | Africa |
| Cape Verde | MT152770.1 | Africa |
| Congo | MT933053.1 | Africa |
| Guinea | MT152763.1 | Africa |
| India | MH639004.1 | Asia |
| Indonesia | OQ891323.1 | Asia |
| Japan | LC546859.1 | Asia |
| Korea | MT641268.1 | Asia |
| Mali | MT152759.1 | Africa |
| Nepal | MT103345.1 | Asia |
| Niger | MT152811.1 | Africa |
| Senegal | MT152792.1 | Africa |
| South Africa | MF593245.1 | Africa |
| Tanzania | MT103348.1 | Africa |
| Taiwan | LC508690.1 | Asia |
| Togo | MT152775.1 | Africa |
| Uganda | MT933059.1 | Africa |
| USA | MK790611.1 | North America |
| Vietnam | MT103337.1 | Asia |
| Zimbabwe | MT103346.1 | Africa |

### 2.2.3. Genetic Population Structure Analyses

Genetic parameters including the number of segregating sites, haplotype numbers, haplotype diversity, nucleotide diversity, theta/site, and Tajima's D [19], were analyzed using DnaSP software v.5.10 [20,21].

### 2.2.4. Genotyping of Isolated *S. frugipedra* by RFLP-PCR

The complete *mt*COXI gene sequence was employed in a PCR-RFLP assay for genotyping. PCR amplification of the full length of COXI was conducted with primers designed based on the complete sequence of the *mt*COXI gene, as documented in Genbank under the entry MN599980.1. Primer design was carried out using the Primer3 program (https://bioinfo.ut.ee/primer3-0.4.0/, accessed on 15 January 2023). The nucleotide sequence of the forward primer, COXI-F, is 5′-ATTCAACAAATCATAAAGATATTGG-3′, and that of the reverse primer, COXI-R, is 5′-AATTCATTATATGAATGTTCAGCTGG-3′. The PCR procedure commenced with an initial denaturation step at 95 °C lasting 3 min and succeeded by 35 cycles consisting of denaturation at 95 °C for 30 s, annealing at 50 °C for 30 s, and elongation at 72 °C for 1 min. A final extension was performed for an extra 10 min at 72 °C. A fragment of DNA of 1500 bp was generated and underwent purification. The digestion was implemented using the restriction enzymes EcoRV, Pst1, and Sac1, which were purchased from Biolabs (Traverse, MI, USA). The digestion was performed following the protocol outlined by the manufacturer. The product of digestion was electrophoresed through 2% agarose gel stained with ethidium bromide, and the molecular length of the digested fragments was identified by photography, using a 100 bp DNA Ladder H3 RTU (GeneDirex, Inc., Las Vegas, NV, USA) The XR+ Gel Documentation System (Bio-Rad, Hercules, CA, USA) was used to photograph the gels. The reference DNA sequence of the *mt*COXI gene, retrieved from entry MN599980.1, was employed to predict the restriction sites using the web of Neb Cutter 2.0 (https://nc3.neb.com/NEBcutter/, accessed on 10 February 2023).

## 3. Results

### 3.1. Molecular Identification of Spodoptera Frugiperda and Construction of Phylogenetic Tree

PCR amplification resulted in a ~700 bp fragment (Figure 3). These fragments corresponded to the COXI gene and were subsequently sequenced. Fragments with identical nucleotide sequences were merged and treated as one sample sequence that represents one specimen. A total of nine sequences, differing by a few bases, were detected and prepared for further analysis. All the obtained mtCOXI sequences were aligned with those deposited in GenBank and scored over 98% identity to mtCOX I of *S. frugiperda* using the BLASTn algorithm. These sequences were then deposited in GenBank and assigned accession numbers, which are presented in Table 2.

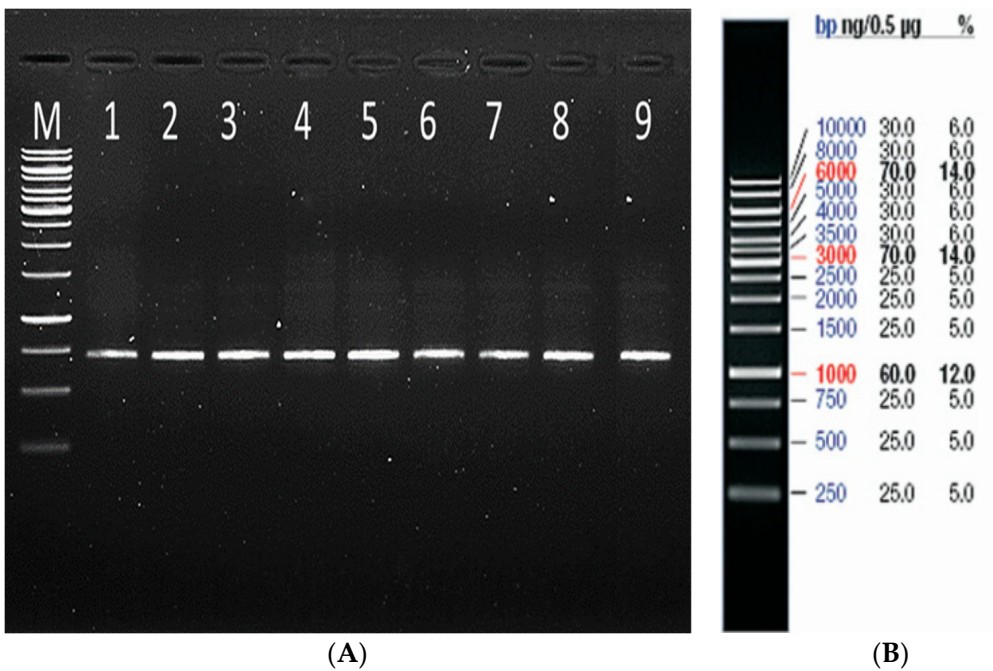

(**A**)                                                                                        (**B**)

**Figure 3.** (**A**) 1.5% agarose gel electrophoresis of mtCOX1 gene; ~700 bp of COX1 gene fragment, M: 1 kb DNA ladder, (**B**) GeneRuler 1 kb DNA ladder (Thermo Scientific™, Waltham, MA, USA).

**Table 2.** Collection details of *S. frugiperda* collected from Egypt governorates and accession numbers assigned to COX1 genes.

| Sites District/City | ID | Samples Coordinate | Accession Number |
|---|---|---|---|
| Shebeen El-koam, Menufia | Shebeen El-koam | 30°33′37.6416″ N, 31°0′28.6128″ E | OQ925919.1 |
| Qillin, Kafr El-sheikh | egy-kafr el-sheikh1 | 31°02′47″ N 30°51′16″ E/31.04639° N 30.854581° E/31.04639; 30.854581 | OP647509.1 |
| Qillin, Kafr El-sheikh | egy-kafr el-sheikh 2 | 26°36′0″ N, 31°39′0″ E | OP649580.1 |
| Qillin, Kafr El-sheikh | Kafr Elsheikh | 26°36′0″ N, 31°39′0″ E | OQ920981.1 |
| Shandawil, Sohag | egy-Sohag | 26°37′11″ N 31°38′46″ E | OP649582.1 |
| Benha, Qalyubia | egy-kaliopia | 30°27′57.5748″ N, 31°11′5.3916″ E | OP648094.1 |
| Kom ombo, Aswan | Aswan1 | 24°28′ N 32°57′ E/24.467° N 32.950° E/24.467 | OQ920901.1 |
| Kom ombo, Aswan | Aswan2 | 24°28′ N 32°57′ E/24.467° N 32.950° E/24.467 | OQ150895.1 |
| Shandawil, Sohag | egy-KRHOSSAM | 26°37′11″ N 31°38′46″ E | OP648105.1 |

Employing the MEGA11 program and the neighbor-joining method analysis with 1000 replicates for bootstrap support, an original tree with well-supported nodes was generated (Figure 4). The analysis revealed two distinct clusters: Cluster I, comprising all isolates in the study except for the isolate egy-KRHOSSAM with accession number OP648105.1, which was included in Cluster II. Phylogenetic tree analysis further demonstrated that

Cluster I included samples from various regions, such as the globally reported rice "R" and corn "C" strains of the FAW. Isolates from Kafr el-Sheikh were confirmed as almost the "R" Rice strain, with accession numbers OP649580 and OP647509, while samples from Aswan, Qalyubia, Menufia, and Sohag were identified as almost the corn "C" strain.

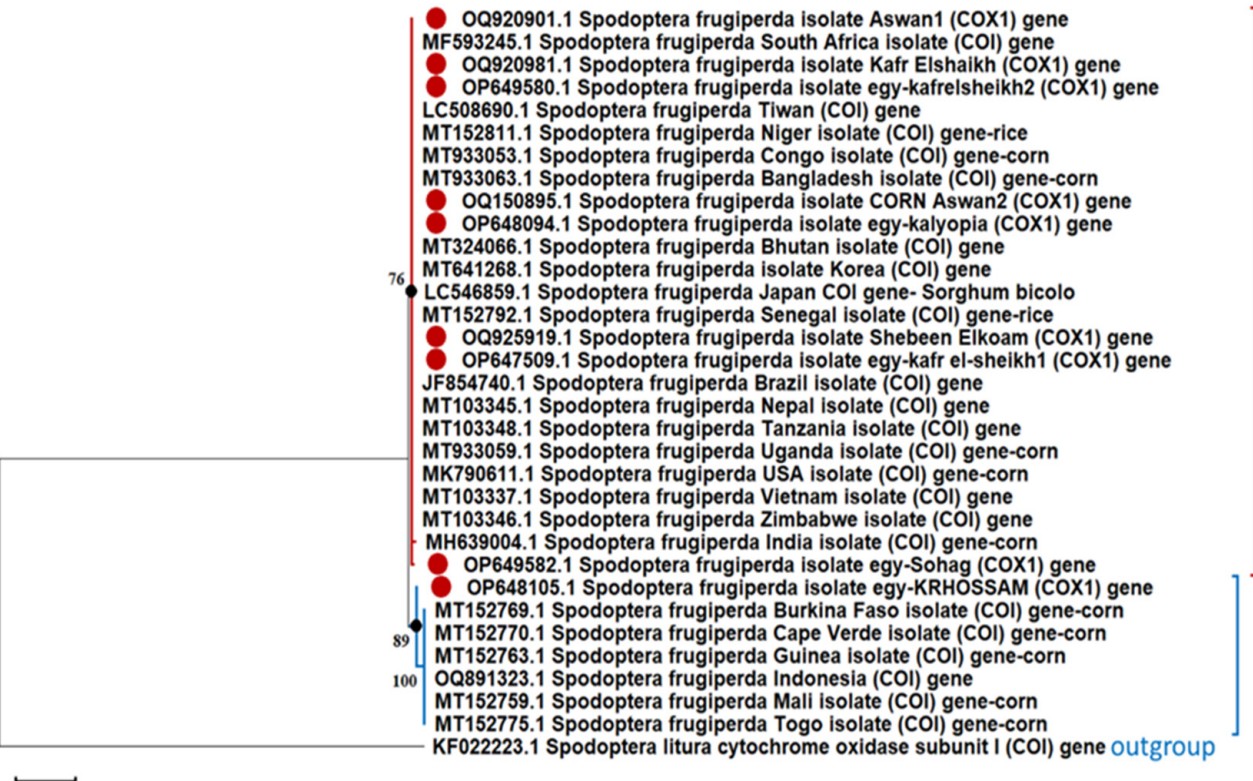

**Figure 4.** Evolutionary relationships among 33 FAW strains utilizing cytochrome oxidase subunit 1 (COX1) sequences; 1000 bootstrap replicates were used to assess robustness using MEGA11. Sequences from the present study are denoted by a red circle on the tree. The red circles represent the insect specimens under study. The red line refers to the cluster I and the blue one refers to the cluster II.

In Cluster II, all samples were of the C-strain, affirming that the strain egy-KRHOSSAM is a corn strain, similar to samples from other countries designated as the "C"-strain. In Cluster I, there were two haplotypes: haplotype 1, which included samples from Shebeen Elkoam, Kafr Elshaikh, Aswan, and Kalyubia, and haplotype 2, which included a population from Sohag. However, in Cluster II, there was one haplotype, haplotype 3, which was represented by egy-KRHOSSAM. The phylogenetic analysis suggests that the most likely native sources of *S. frugiperda* in Egypt are Asia and Africa.

The nucleotide sequences of COXI investigated in our study were aligned against those of the rice and corn reference strains, isolated from Burkina Faso and Niger, respectively. The analysis in Table 3 revealed nucleotide variations at positions 105, 195, 246, 477, and 558. Based on this analysis, the strains obtained from Kafr el-Sheikh (accession numbers OP647509 and OP649580) were probably designated as "R". However, the laboratory strain egy-KRHOSSAM exhibited the same nucleotide variations as in the corn strain from Burkina Faso at these five positions, confirming its classification as a corn strain.

**Table 3.** Comparison of nucleotide variations in COX1 Genes of *S. frugiperda* strains: current investigation vs. previously documented Rice and Corn strains, emphasizing nucleotides polymorphism. Rows with the same background color indicate identical nucleotides in corresponding positions, whereas a colorless background denotes differing nucleotides in those positions.

| Accession Numbers, Strains | Nucleotide Position | | | | |
|---|---|---|---|---|---|
| | **105** | **195** | **246** | **477** | **558** |
| OP648105, egy-KRHOSSAM (**laboratory strain—this study**) | G | T | C | T | C |
| **MT152769, *S. frugiperda*, Burkina Faso (Corn strain)** | **G** | **T** | **C** | **T** | **C** |
| OP647509, egy-kafr el-sheikh1 (**this study**) | A | A | T | C | T |
| OP649580, egy-kafr el-sheikh 2 (**this study**) | A | A | T | C | T |
| OQ920981, Kafr Elshaikh (**this study**) | A | A | T | T | C |
| OQ925919, Shebeen El-koam (**this study**) | G | T | T | T | C |
| OP649582, egy-Sohag (**this study**) | G | T | C | T | C |
| OP648094, egy-kaliopia (**this study**) | G | T | C | T | C |
| OQ920901, Aswan1 (**this study**) | G | A | C | T | C |
| OQ150895, Aswan2 (**this study**) | G | A | T | T | C |
| **MT152811- *S. frugiperda*, Niger (Rice strain)** | **A** | **A** | **T** | **C** | **T** |

### 3.2. Polymorphism Results

The haplotype diversity of *S. frugiperda* in Egypt was investigated by analyzing a 567 bp segment of the mtCOX I barcode region obtained from populations sampled across five governorates. An analysis revealed the presence of five polymorphic sites, with a calculated nucleotide diversity of 0.00258. Three distinct haplotypes were identified from the COX I sequences, exhibiting a haplotype diversity (Hd) of 0.392, with a variance of haplotype diversity estimated at 0.01759. The majority of COI sequences obtained in this study shared a common haplotype (H1), comprising populations from Shebeen Elkoam, Kafr Elshaikh, Aswan, and Kaluyopia. However, the population from Sohag (Accession no. OP649582) exhibited a second haplotype (H2), while a third haplotype (H3) was observed in the KRHOSSAM population (Accession no. OP648105). Fu's Fs (2.074) and Fu and Li's D (1.30779) values were both positive, indicating an excess of rare haplotypes, whereas Tajima's D was negative (−0.95225), suggesting an excess of intermediate-frequency alleles within the populations studied. Furthermore, nine haplotypes were identified among the twenty-three populations analyzed, representing the entries obtained from GenBank. The populations from the USA and Asia displayed the highest levels of haplotype and nucleotide diversities, whereas the populations from Africa exhibited the lowest levels. All these results are illustrated in Table 4.

**Table 4.** Polymorphic sites, nucleotide, and haplotype diversity of *S. frugiperda* in Egypt.

| Number of Polymorphic Sites | Nucleotide Diversity | Haplotype Diversity | Variance of Haplotype Diversity | Fu's Fs | Fu and Li's D | Tajima's D |
|---|---|---|---|---|---|---|
| 5 | 0.00258 | 0.392 | 0.01759 | 2.074 | 1.30779 | −0.95225 |

### 3.3. Analysis of PCR-RFLP Results

The total length of 1500 bp for COXI was generated by PCR using COXI-F and COXI-R primers. These fragments from the samples under investigation were digested by EcoRV, Pst1, and Sac1, resulting in the appearance of two patterns. The first pattern appears in Figure 5A; three bands at molecular weights of 250 bp, 700 bp, and 1500 bp were generated after digestion by EcoRV in cases of samples egy-kafr el-sSheikh1 (accession no. OP647509 and egy-kafr el-sheikh 2 accession no. OP649580). However, in the other samples, the pattern of electrophoresis was the same, where bands of 1500 bp, 900 bp, and 700 bp appeared. The second pattern appeared after digestion by the restriction enzyme Pst1 Figure 5B. Two bands of 400 bp and 600 bp were generated in egy-kafr el-sheikh1 and egy-kafr el-sheikh 2; however, the digestion in other samples did not occur. That is because

there were no specific restriction sites for the Pst1 restriction enzyme. Figure 5C illustrates that the pattern of digestion by Sac1 was the same for all samples in this study. These results supposed that egy-kafr el-sheikh1, and egy-kafr el-sheikh 2 belong to the same haplotype and are almost all rice strains, while the others are almost corn strains. We can conclude that the restriction enzymes Pst1 and EcoRV are effective tools for distinguishing the genotypes involved in this study.

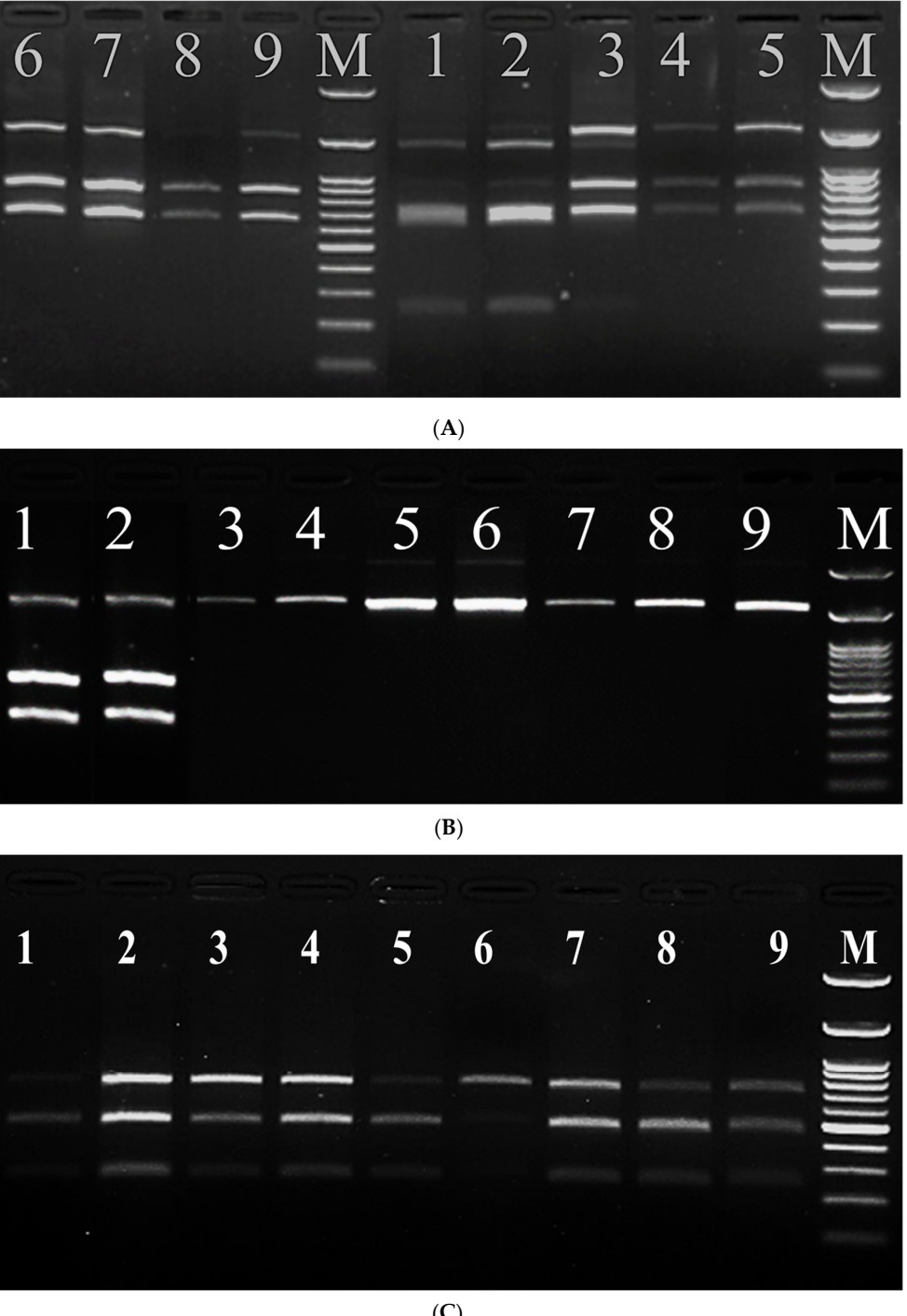

**Figure 5.** PCR-RFLP pattern of the mitochondrial COX I gene (complete sequence) digested by restriction enzymes: (**A**) EcoRv; (**B**) Pst1; (**C**) Sac1. Lanes 1 and 2 represent populations with accession numbers OP647509 and OP649580, respectively. Lanes 3–9 represent populations with accession numbers OQ925919, OQ920981, OP649582, OP648094, OQ920901, OQ150895, and OP648105. M: 100 bp DNA ladder H3 RTU (GeneDirex, Inc., Las Vegas, NV, USA).

## 4. Discussion

A suitable strategy for investigating population genomics involves comprehensive monitoring of a population's evolutionary path over a specific geographic area or time span. *Spodoptera frugiperda*, due to its recent range expansion, serves as an excellent model for such studies. This species has been extensively studied for its diversity in both its original and expanded habitats [22]. Hence, understanding the diversity of the Egyptian fall armyworm (FAW) population was deemed crucial following its widespread migration across Egypt. In this study, we genetically characterized fall armyworm (FAW) specimens collected from cornfields across five governorates using molecular markers targeting the COX gene. The mitochondrial COX gene contains variations that are highly useful for studying fall armyworm populations. Specifically, the 5′ segment of the COX locus, referred to as COXA, contains the barcode region, which plays a key role in identifying both species and host strains [23]. On the other hand, the COXB segment, located closer to the 3′ end, generates haplotypes that are essential for distinguishing between strains and differentiating populations that are geographically distant from one another [24]. In our study, we used the 3′ end corresponding to COXB. However, upon initial examination of strain-defining loci within the COXI gene, we noted limited variation in our dataset. Surprisingly, both Rice and Corn strains were identified in specimens collected from corn fields across Egyptian governorates, despite the insect's well-known preference for its host plant. These results indicate a potential discrepancy between the COI marker and the actual host plants of the fall armyworm. This inconsistency may arise from the insect's adaptable nature in selecting host plants or from limitations in the marker's ability to accurately differentiate between the corn and rice strains of the fall armyworm. These findings align with similar studies conducted previously, underscoring the importance of further research to elucidate the underlying factors contributing to the observed genetic variations in fall armyworm populations [8,25].

In this investigation, samples from Kafr el-Sheikh displayed two genotypes: egy-Kafr el-Sheikh1 (OP647509.1) and egy-Kafr el-Sheikh 2 (OP649580.1), designated as the Rice strain. However, the strain from Kafr Elshaikh (OQ920981.1) and other specimens from different governorates were identified as the Corn strain. The coexistence of both strains (Rice and Corn) in Kafr el-Sheikh can be attributed to the governorate's focus on both rice and corn cultivation. During the collection in Kafr el-Sheikh, where two rice and one corn specimen were recovered, the rice specimens were found in corn fields (during early-summer corn growth) despite nearby rice fields being free of *S. frugipedra* infestation. This behavior can be attributed to the rough surface of rice-plant leaves during that period, which hindered larvae feeding and likely prompted the insect to transfer to maize plants. In contrast, other studied regions primarily cultivated corn, making it the main host for *Spodoptera frugiperda* [5,26–28]. Gene sequencing of COXI in these samples identified three distinct haplotypes, with the Corn strain being the most prevalent. Concerning sample "egy KRKOSSAM", it was initially obtained from a corn field in Aswan and subsequently propagated for twelve generations under laboratory conditions. Its clustering with corn strains provided assurance of minimal gene flow and consistent genetic homogeneity compared to other samples collected from corn fields, which have undergone substantial hybridization and genetic modification processes. Our results corroborate that the larvae and haplotypes of the R-strain were found to primarily originate from maize rather than rice, indicating a tendency for multiple host-plant choices. This observation potentially explains the occurrence of rare R-strain haplotypes in our dataset, derived from both our sampling efforts and sequences obtained from GenBank. It appears that there is no fixed association between a specific strain and its preferred host plant, nor is there a consistent geographical pattern. Instead, the species exhibits variations both within and between strains, likely influenced by factors such as the agroecosystem, geographical location, and the specific parameters of each study. The appearance of different haplotypes in Egypt highlights the impact of regional influences on shaping genetic diversity [29]. Genetic drift and natural selection are identified as primary factors governing molecular-level changes

within and among insect populations [30]. The investigation of the current population's genetic structure as an invasive species aids in identifying the source of infestation, invasion paths, and genetic consequences. The invasion process typically occurs in three distinct phases [31]. First, it involves the introduction of a non-native species to an area outside of its original habitat, often facilitated by human activities such as trade, urbanization, and infrastructure development [32]. The second phase entails the establishment of stable populations, supported by either a higher intrinsic growth rate [33] or superior competitive ability compared to native species [34]. Subsequently, in the third phase, the species expands its range from these established populations. Adaptive evolution influences each stage, enabling the fixation of advantageous mutations or genetic variations, facilitating hybridization between introduced and native species and potentially involving genome doubling. This process is often characterized by a temporal gap known as the 'lag phase' between introduction and range expansion [35]. Identifying alien species before they become invasive is crucial for effective pest management, as eradication efforts are typically only feasible during this lag phase. This can be translated by investigating the genetic studies and the phylogenetic analysis that provide essential tools for understanding the origin and adaptive strategies of *S. frugiperda* in new environments [36].

In this study, we utilized genetic studies and phylogenetic analysis as essential tools to unravel the origin of *S. frugiperda* and to comprehend how these organisms adapt to and thrive in new environments. The analysis of population genetics based on the COXI gene revealed that *S. frugiperda* populations in Egypt are undergoing evolution in a neutral pattern. This is substantiated by the disappearance of novel haplotypes and the observed low genetic diversity in populations, contrasting with the expanding pattern observed in American and Asian populations. These findings align with conclusions drawn in other studies, such as those by Nayyar et al. [37] and Omuut et al. [8], that reported that *S. frugiperda* populations in America and Africa are also evolving neutrally and expanding. Regarding the genetic variation, our findings align with previous studies, which, despite being conducted in different locations, drew similar conclusions (34, 79, 133, 169, 220, 451, 526, 532, 562, 596, and 625). Discrepancies in specific positions may be attributed to variations in the amplified sequence size, as reported in various studies [38–40].

It is crucial to note that the association of Corn and Rice strains with host plants is not exclusive, as the COXI gene marker was detected in both strains collected from corn and rice fields [41,42]. This suggests the inability of the COXI gene marker to discriminate between the two FAW strains, possibly due to the plasticity in plant choice by *S. frugiperda* [30,43–45]. Nevertheless, multiple genetic studies have proposed that invasive fall armyworms (FAWs) are hybrids of the Corn and Rice strains [23]. Research utilizing marker-based analyses indicates that the majority of invasive FAWs exhibit genetic traits of both Corn and Rice, implying that interstrain hybridization has played a role in their development [43,44]. Nevertheless, it is still unclear whether this pattern persists when examining whole-genome sequences. In this study, we utilized an additional technique to reinforce the findings obtained from the analysis of genetic diversity in fall armyworm (FAW) insects, specifically through the sequencing of the COX gene. This technique includes Top of Form PCR-RFLP based on the whole nucleotide sequence of the COXI gene. In a comparable investigation conducted by Hazel et al. [46], the utilization of MspI restriction digestion on PCR products successfully distinguished DNA extracts originating from Corn and Rice strains of *S. frugiperda*. Notably, MspI digestion yielded the anticipated fragment sizes (497 and 72 bp) for the *S. frugiperda* Corn strain in PCR products obtained from corn-fed larvae, colony-reared insects, and sorghum larvae, as well as Sf9 cells. However, it proved ineffective in digesting PCR products derived from rice- or bermudagrass-fed larvae. Another study proved the efficacy of the restriction enzymes Dra I, Alu I, and Nla III to distinguish the seven noctuid species [47]. Here, The PCR-RFLP technique offers a significant advantage in that it eliminates the possibility of false negatives, as the restriction patterns are specific to DNA sequences.

Previous studies conducted by researchers in the United States [48] and Colombia [49,50] have utilized AFLP molecular markers to differentiate between the larvae and adults of both *S. frugiperda* strains. However, AFLP markers were found to be less precise compared to the RFLP markers employed in our study, as they produced numerous polymorphisms, complicating the assignment to each strain. AFLP was standardized for Corn and Rice strain identification in the United States and concluded that these markers effectively differentiated both strains based on their band patterns [48]. Similarly, when these primers were applied to *S. frugiperda* larvae collected in Colombia, they did not discern specific band patterns for each strain [49,50]. In sum, despite AFLP being considered suitable markers for population genetics studies, Lobo-Hernández and Saldamando-Benjumea [48] acknowledged their inability to effectively differentiate between the two strains.

## 5. Conclusions

Our research represents the inaugural attempt to reveal the genetic diversity of the fall armyworm across its entire global distribution, especially after its recent invasions into various regions. The gathered data indicate that Egypt is potentially emerging as a significant hotspot for the proliferation of FAW, particularly with regard to the COXI-A corn haplotypes. Additionally, our findings highlight the occurrence of inter-strain recombination in India, suggesting the potential for an evolution. Given that FAW has expanded its host range in Egypt, our study contributes valuable insights into establishing the connection between the host range and the novel FAW haplotypes identified in Egypt. Subsequent investigations should focus on evaluating the effectiveness of different insecticides and biocontrol agents against the diverse FAW haplotypes present in Egypt. In conclusion, this study not only serves as a foundational reference for understanding the current genetic landscape of FAW in Egypt but also provides a basis for monitoring its future evolution and spread within Egypt and across its entire demographic range.

**Author Contributions:** Conceptualization, K.A.E.L., R.G.S., G.M.E.-S. and S.H.M.; methodology, G.M.E.-S., R.G.S. and S.H.M.; software, G.M.E.-S., R.G.S. and S.H.M.; validation, G.M.E.-S., R.G.S., K.A.E.L. and S.H.M.; formal analysis, G.M.E.-S., R.G.S. and S.H.M.; investigation, G.M.E.-S., R.G.S., K.A.E.L. and S.H.M.; resources, G.M.E.-S. and R.G.S.; data curation, G.M.E.-S., R.G.S. and S.H.M.; writing—original draft preparation, G.M.E.-S. and S.H.M.; writing—review and editing, G.M.E.-S. and S.H.M.; visualization, G.M.E.-S., R.G.S. and S.H.M.; supervision, K.A.E.L.; project administration, G.M.E.-S., R.G.S., K.A.E.L. and S.H.M.; funding acquisition, G.M.E.-S., R.G.S., K.A.E.L. and S.H.M. All authors have read and agreed to the published version of the manuscript.

**Funding:** This research received no external funding.

**Data Availability Statement:** All data are available in the manuscript.

**Acknowledgments:** We extend our thanks to the National Research Centre for providing us with access to facilities, necessary chemicals, and instruments essential for the advancement of our research. We extend our gratitude to Hosam Mohamed Khalil Hammam ElGepaly at the Shandaweel Research Station at the Agriculture Research Centre for kindly providing us with the KRHOSSAM strain.

**Conflicts of Interest:** The authors declare no conflicts of interest.

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
