# Peer review of "Identification and Genetic Diversity of Spodoptera frugiperda J. E. Smith (Lepidoptera: Noctuidae) in Egypt"

_agronomy, doi:10.3390/agronomy14040809_

Round 1

Reviewer 1 Report

Comments and Suggestions for Authors

The document "Identification and Genetic Diversity of Spodoptera frugiperda J. E. Smith (Lepidoptera: Noctuidae) in Egypt" is well written, contains outstanding information on the diversity of S. frujiperda and its ability to invade new crops and adapt to achieve partially isolated populations since the genetic point of view.

Some specific observations are in the pdf.

In general terms there are three observations:

Introduction

1.- The locations of Egypt mentioned should be located where they are, in the north, south, east, and west, so that whoever reads it can locate themselves more easily. As defined, all the villages are in the Upper Egypt.

Materials and Methods

Improve the quality of figure 2, locate the regions according to the terms used, upper and lower Egypt.

2. Explain why a laboratory colony called HOSSAM was established, what was the objective, and whether was it only used as a control??? to explain what?

The authors are asked for this explanation because the document describes comparing the R and C strains.

3. Results

Discussion

The discussion is generally fine, but the focus according to the title is to differentiate the different strains, and it is argued that they infest different crops and are the factor of their specialization. Then, the results should be discussed in terms of what the genetic differences between S. frugiperda populations mean the ability to invade and specialize in new crops, and how they differentiate in isolated populations or maintain genetic exchange.

Author Response

Response letter

March / 31 /2024

Manuscript ID (agronomy-2915448 )

Dear Reviewer

We thank you for your valuable comments regarding our article entitled, " Identification and Genetic Diversity of Spodoptera frugiperda J. E. Smith (Lepidoptera: Noctuidae) in Egypt". We agree with all comments raised and found them very helpful. we would like to thank you for your time and efforts necessary to provide such insightful guidance. Yellow highlight was used to mark the revision. Below, we address each criticism and comments individually, and explain how we have modified the manuscript to address the concerns that were expressed. We kindly ask that you consider the revised article for publication.

Introduction

1- The locations of Egypt mentioned should be located where they are, in the north, south, east, and west, so that whoever reads it can locate them more easily. As defined, all the villages are in the Upper Egypt.

Response: Thank you for bringing this to our attention. The locations have now been clarified.

Materials and Methods

Improve the quality of figure 2, locate the regions according to the terms used, upper and Lower Egypt.

Response: The image resolution has been improved to 300 dpi. Additionally, the regions' locations have been clearly indicated both in the text and the figure title. We hope this enhanced clarity meets your requirements. 

  1. Explain why a laboratory colony called HOSSAM was established, what was the objective, and whether was it only used as a control??? to explain what? The authors are asked for this explanation because the document describes comparing the R and C strains.

Response: The process of establishing this strain has been detailed in the introduction and materials sections. We trust this aligns with your expectations. 

  1. Results

Discussion

The discussion is generally fine, but the focus according to the title is to differentiate the different strains, and it is argued that they infest different crops and are the factor of their specialization. Then, the results should be discussed in terms of what the genetic differences between S. frugiperda populations mean the ability to invade and specialize in new crops, and how they differentiate in isolated populations or maintain genetic exchange.

Response: Genetic studies examining this topic have presented varied perspectives. This manuscript's discussion aims to highlight these differing viewpoints alongside the authors' interpretation. These points have been incorporated and emphasized within the discussion sections.

Reviewer 2 Report

Comments and Suggestions for Authors

The authors collected five populations of Spodoptera frugiperda in five provinces of Egypt. Then the genetic makeup was analyzed by cloning and phylogenetic analysis of partial sequences of the mitochondrial cytochrome oxidase subunit I (COI). Furthermore, the genotype patterns of the populations were analyzed by PCR-RFLP. This is an interesting work. While some comments need to be responded to.

1.     In the introduction, the Latin literary name Spodoptera frugiperda should be preceded by full names at first place, followed by an abbreviation.

2.     The five provinces that collected the insects should be marked in Figure 2.

3.     Why the authors only used mtCOXI genes for the analysis in this study? Many reports also use Triose-phosphate isomerase (Tpi) or other genes together with COXI genes for the analysis.

4.     The authors collected the insects from corn crops, while they identified that egy-kafr el-sheikh1 and egy-kafr el-sheikh 2 belong to the same haplotype and are almost all rice strains. The authors need to discuss the reason.

5.     There are many format errors in the manuscript; please check them carefully. For example, S. frugiperda need to be italic.

Comments on the Quality of English Language

No

Author Response

Response letter

March / 31 /2024

Manuscript ID (agronomy-2915448 )

Dear Reviewer

We thank you for your valuable comments regarding our article entitled, " Identification and Genetic Diversity of Spodoptera frugiperda J. E. Smith (Lepidoptera: Noctuidae) in Egypt". We agree with all comments raised and found them very helpful. we would like to thank you for your time and efforts necessary to provide such insightful guidance. Yellow highlight was used to mark the revision. Below, we address each criticism and comments individually, and explain how we have modified the manuscript to address the concerns that were expressed. We kindly ask that you consider the revised article for publication.

Comments and Suggestions for Authors

The authors collected five populations of Spodoptera frugiperda in five provinces of Egypt. Then the genetic makeup was analyzed by cloning and phylogenetic analysis of partial sequences of the mitochondrial cytochrome oxidase subunit I (COI). Furthermore, the genotype patterns of the populations were analyzed by PCR-RFLP. This is an interesting work. While some comments need to be responded to.

  1. In the introduction, the Latin literary name Spodoptera frugiperdashould be preceded by full names at first place, followed by an abbreviation.

Response

Thank you for your comment. Insect name was corrected.

  1. The five provinces that collected the insects should be marked in Figure.

Response

Thank you for your comment. Provinces names were marked accordingly.

  1. Why the authors only used mtCOXI genes for the analysis in this study? Many reports also use Triose-phosphate isomerase (Tpi) or other genes together with COXI genes for the analysis.

Response

 The current study aimed to intially confirm invasion of Spodoptera frugiperda to Egypt by basic and standard molecular identification technique and screen haplotype number for further tracking demographic events. This method of identifcation is robust and widely used for identification and gentic population structure studies untill now (fore example Malekera et al., 2023). Thus the aims suits methodology.

In our prspective study, we plan to extend our studies and monitor the expansion of the pest all over Egypt with more specific marker.

Malekera, M.J.; Mamba, D.M.; Bushabu, G.B.; Murhula, J.C.; Hwang, H.-S.; Lee, K.-Y. Genetic Diversity of the Fall Armyworm Spodoptera frugiperda (J.E. Smith) in the Democratic Republic of the Congo. Agronomy 2023, 13, 2175. https://doi.org/10.3390/ agronomy13082175 

  1. The authors collected the insects from corn crops, while they identified that egy-kafr el-sheikh1 and egy-kafr el-sheikh 2 belong to the same haplotype and are almost all rice strains. The authors need to discuss the reason.

Response

Being the two locations are gathered in one haplotype, this is not necessarily reflecting their feeding strain while reflecting their inheritance from single parent and the same allele.

  1. There are many format errors in the manuscript; please check them carefully. For example, S. frugiperda need to be italic.

Response

Insect name was corrected through the whole manuscript.

Reviewer 3 Report

Comments and Suggestions for Authors

This study analyze the genetic makeup of the fall armyworm population by collecting specimens from five provinces across Egypt. Through examination of partial sequences of the mitochondrial cytochrome oxidase subunit I (COI) and employing the PCR-RFLP technique on the complete COI sequence, this research contributes essential insights into the genetic diversity of insects in Egypt. And I quite approve of this work, but this paper is not up to the standard of being published yet, which requires a great of work by the authors to modify, detailed modification suggestions are as follows:

Introduction:

1.     Line35-36: “southern and eastern Africa” = “Southern and Eastern Africa”

2.     Line46: The genus name of Spodoptera frugiperda should be abbreviated.

3.     Line70-71: The article an may be incorrect. It should be “a precise technique”

4.     Line73: S. Frugiperda” = “S. frugiperda

Materials and Methods:

1.     Line102: “pre-serving” = “preserving”

2.     Line138: ……S. litura as outgroup” = “……Spodoptera litura as a outgroup”.

3.     Line154: “S. frugipedra” = “S. frugipedra

4.     Line164: The word “a” should be capitalized in this context.

5.     Line170: There should be a space after “using”.

Results:

1.     Figure 3 and Figure 5 should indicate the size of bands of the DNA ladder.

2.     Line185,223: “S. frugipedra” = “S. frugipedra

3.     Please illustrate the ‘3.2. Polymorphism results” in a graph or table.

4.     Line212: “The nucleotides sequences” = “The nucleotide sequences”

5.     Line258: “Lanes 1 and 2 represents populations” = “Lanes 1 and 2 represent populations”

Discussion:

1.     Line335: It seems that quantifier use may be incorrect here. It should be “Another study”.

References:

1.      Line392, 398, 404, 407: “Spodoptera frugiperda” = “Spodoptera frugiperda”. Please correct similar problems in the literature

2.      Please correct the format of reference13.

Comments on the Quality of English Language

Minor editing of English language required.

Author Response

Response letter

March / 31 /2024

Manuscript ID (agronomy-2915448 )

Dear Reviewer

We thank you for your valuable comments regarding our article entitled, " Identification and Genetic Diversity of Spodoptera frugiperda J. E. Smith (Lepidoptera: Noctuidae) in Egypt". We agree with all comments raised and found them very helpful. we would like to thank you for your time and efforts necessary to provide such insightful guidance. Yellow highlight was used to mark the revision. Below, we address each criticism and comments individually, and explain how we have modified the manuscript to address the concerns that were expressed. We kindly ask that you consider the revised article for publication.

This study analyze the genetic makeup of the fall armyworm population by collecting specimens from five provinces across Egypt. Through examination of partial sequences of the mitochondrial cytochrome oxidase subunit I (COI) and employing the PCR-RFLP technique on the complete COI sequence, this research contributes essential insights into the genetic diversity of insects in Egypt. And I quite approve of this work, but this paper is not up to the standard of being published yet, which requires a great of work by the authors to modify, detailed modification suggestions are as follows:

Introduction:

  1. Line35-36: “southern and eastern Africa” = “Southern and Eastern Africa”

Response: done

  1. Line46: The genus name of Spodoptera frugiperdashould be abbreviated.

Response: done

  1. Line70-71: The article an may be incorrect. It should be “a precise technique”

Response: done

  1. Line73: “ Frugiperda” = “S. frugiperda

Response: done

Materials and Methods:

  1. Line102: “pre-serving” = “preserving”

Response: done

  1. Line138: “……litura as outgroup” = “……Spodoptera litura as a outgroup”.

Response: done

  1. Line154: “S. frugipedra” = “ frugipedra

Response: done

  1. Line164: The word “a” should be capitalized in this context.

Response: done

  1. Line170: There should be a space after “using”.

 Response: done

Results:

  1. Figure 3 and Figure 5 should indicate the size of bands of the DNA ladder.

 Response:100 base pair ladder was used with band size 100,200,300,400,500,600,700,800, 900, 1000 and 1500 bp and additional DNA ladder figure has been incorporated into the figures.

  1. Line185,223: “S. frugipedra” = “ frugipedra

Response: done

  1. Please illustrate the ‘3.2. Polymorphism results” in a graph or table.

Response:

  1. Line212: “The nucleotides sequences” = “The nucleotide sequences”

Response: done

  1. Line258: “Lanes 1 and 2 represents populations” = “Lanes 1 and 2 represent populations”

Response: done

Discussion:

  1. Line335: It seems that quantifier use may be incorrect here. It should be “Another study”.

Response: done

References:

  1. Line392, 398, 404, 407: “Spodoptera frugiperda” = “Spodoptera frugiperda”. Please correct similar problems in the literature

 Response: done

  1. Please correct the format of reference13.

Response: done
